# Antimicrobial Compounds Isolated from Endolichenic Fungi: A Review

**DOI:** 10.3390/molecules26133901

**Published:** 2021-06-25

**Authors:** A. Nethma Wethalawe, Y. Vindula Alwis, Dinusha N. Udukala, Priyani A. Paranagama

**Affiliations:** 1Institute of Chemistry Ceylon, College of Chemical Sciences, Rajagiriya 10100, Sri Lanka; nethma@ichemc.edu.lk (A.N.W.); vindula@ichemc.edu.lk (Y.V.A.); dinusha@ichemc.edu.lk (D.N.U.); 2Department of Chemistry, University of Kelaniya, Kelaniya 11600, Sri Lanka

**Keywords:** endolichenic fungi, antibacterial, antifungal, antiviral, antiplasmodial, secondary metabolites

## Abstract

A lichen is a symbiotic relationship between a fungus and a photosynthetic organism, which is algae or cyanobacteria. Endolichenic fungi are a group of microfungi that resides asymptomatically within the thalli of lichens. Endolichenic fungi can be recognized as luxuriant metabolic artists that produce propitious bioactive secondary metabolites. More than any other time, there is a worldwide search for new antibiotics due to the alarming increase in microbial resistance against the currently available therapeutics. Even though a few antimicrobial compounds have been isolated from endolichenic fungi, most of them have moderate activities, implying the need for further structural optimizations. Recognizing this timely need and the significance of endolichenic fungi as a promising source of antimicrobial compounds, the activity, sources and the structures of 31 antibacterial compounds, 58 antifungal compounds, two antiviral compounds and one antiplasmodial (antimalarial) compound are summarized in this review. In addition, an overview of the common scaffolds and structural features leading to the corresponding antimicrobial properties is provided as an aid for future studies. The current challenges and major drawbacks of research related to endolichenic fungi and the remedies for them have been suggested.

## 1. Introduction

Thousands of microorganisms, including fungi and bacteria, often associate with living and dead plant tissues [1]. Oftentimes, the importance of these microorganisms is unobserved; only the saprotrophic and pathogenic relationships are being investigated, and are viewed as a troublesome group of organisms. However, there are groups of micro-organisms that are phyto-friendly and able to produce a plethora of secondary metabolites with significant biological activities that will aid them to adapt better to their surroundings [2,3].

Lichens are an amalgamation or a symbiotic partnership between a fungus and a photosynthetic organism. The heterotrophic fungal partner is termed as a “mycobiont”, while the photoautotrophic “photobiont”, could be either green algae or cyanobacteria [4,5,6]. As the mycobiont usually plays the more prominent member, lichens have traditionally exhibited characteristics similar to theirs. The two partners share water, nutrients and gases [6] and this mutualism allows the lichens to develop under extremely exceptional ecological conditions like deserts, rocky coasts, alpine zones and droughts [5,6,7,8]. Living under these unusual conditions enables lichens to give birth to a variety of luxurious compounds with complex structures and numerous bioactivities, making this a highly interesting stream for natural product chemists to pursue [8,9,10]. However, lichen flora is less abundant in the neighbourhood of urban and industrialized areas, as lichens are easily affected by air pollutants [11]. Lichens display a wide distribution of more than 20,000 different species worldwide [12], and have been utilized on various occasions in the past, such as in food, perfumes, dyes and as antidotes for folk medication [10,11].

The mycobiont reportedly produces around 1000 chemically variegated lichen substances [13], most of which are specific to lichens, with only a small amount occurring in other fungi, algae and higher plants [14]. These metabolites were found to showcase significant bioactivities including antibacterial [15], antifungal [16], anti-inflammatory [17], etc. However, as they grow slowly in nature, their applications are limited [18]. Moreover, over-collecting lichens negatively affect their conservation. Consequently, there arises a need to prioritize other fast-growing organisms that can be cultured easily. Incidentally, the discovery of another group of asymptomatic fungi that are analogous to endophytic fungi, flourishing inside the tissues of healthy lichen thalli, came about [19,20]. This group of micro-fungi, called endolichenic fungi (ELF), was first explained in 1990 [21]. The different types of ELF mainly consist of Ascomycetes, Basidiomycetes, Coelomycetes and Hyphomycetes [6]. They resemble endophytic fungi in many aspects; they (i) live in the intercellular spaces of the hosts, (ii) do not show any noticeable disease symptoms, (iii) are transferred horizontally and (iv) give rise to a mixture of bioactive secondary metabolites [19,22,23]. However, the ELF are different to mycobionts [24], which form the lichen thallus, and from lichenicolous fungi—an ecological group of mitosporic and meiosporic fungi that can often be seen on living lichens [19]. This provides support to the knowledge on ELF consisting of secondary metabolites that are speculated to be different from those generated by lichens [25]. These substances show numerous structural scaffolds, including alkaloids, terpenes, quinones, aromatic compounds, peptides, oxygenated heterocycles, steroids, and allylic compounds were also found to display many different biological activities [26,27,28,29,30].

The part played by ELF on lichen symbiosis has not been fully studied, although some assumptions have been put forward. For example, the occurrence of ELF is thought to make an important ecological contribution towards lichens by helping their growth and formation, as well as by safeguarding the host against insect herbivores [31]. The contrast between bioactivities of ELF and their host lichens are assisted by the revelation of a distinct batch of metabolites generated by these organisms, as uncovered by metabolomics analyses. ELF are believed to contribute to lichen association by supplying different metabolites that could inhibit the growth of “competitor” microbes and other harmful invaders. The same has been noticed for endophytic fungi in plants, even though the host plants and their associated endophytes give rise to similar secondary metabolites [32]. Culturing ELF in the presence of their host lichen thallus would probably unearth comparable metabolites and would help widen the span of metabolic pathways and products of both parties [33].

ELF nurtures a wealthy source of bioactive natural product metabolites with wide-ranging biological activities. Depending on the unique habitats of lichens, the residing ELF might lead to fascinating metabolites. The chemical composition and bioactivity of a large number of ELF have not been studied completely, leaving behind an enormous library of compounds to be investigated [6]. A huge amount of effort awaits to uncover their potential as producers of natural products. If the constraints of current procedures and technologies could be overcome, a new era could begin for natural products emerging from them for the well-being of human health [34].

## 2. Antimicrobial Compounds Extracted from Endolichenic Fungi

The need for new antimicrobial drugs is enhanced by the emergence of microbial resistance against almost all the currently available antibiotics and the sudden appearance of deadly viral infections [35]. Discovery of novel antimicrobial drugs was speculated as a solution to the growing threat of antibiotic-resistant microorganisms by the former secretary general of the United Nations, Ban Ki-Moon, at the UN General Assembly in 2016 [36].

The significant role of fungal species in producing antibiotics is elicited after the discovery of Penicillin G in 1928 [7]. Symbiotic fungal species like endophytic fungi are known to produce a plethora of antimicrobial compounds pertinent in therapeutics and agriculture [37]. Similar to other symbiotic fungi, ELF produces several secondary metabolites, which protect the lichen from biotic as well as abiotic stress [38]. ELF-derived antimicrobial compounds are one such group of metabolites essential to overcome the constant microbial threats faced by lichens. In some cases, extracts or natural products isolated from ELF show strong antimicrobial properties even though these bioactivities are not naturally observed within their ecological niche.

The discovery of many antimicrobial metabolites from ELF, establishes a hopeful satisfaction to the perpetual thirst for new antimicrobial drugs. However, these compounds might need further optimizations to modify their pharmacological and toxicological profiles. On the other hand, the development of synthetic pathways to produce these compounds, an industrial scale is essential to minimize the cost of production and minimize the environmental impacts. Thus, a detailed summary is provided here to describe the antimicrobial compounds isolated from ELF, including their sources, structures, activities and potencies in antimicrobial drug discovery.

Addressing the aforesaid requirement, Table 1 of this review provides an overview of antimicrobial secondary metabolites isolated from ELF, which includes 31 antibacterial compounds, 58 antifungal compounds, two antiviral compounds, and one antiplasmodial (antimalarial) compound. The structures of these compounds are given in the Figure 1. Most of the authors have either reported only the antimicrobial properties of the lichenic and endolichenic fungal extracts without isolation of the metabolites responsible for the relevant bioactivity or have not quantified it in the form of Minimum Inhibitory Concentration (MIC) or IC_50_. However, only the endolichenic fungal secondary metabolites, whose antimicrobial properties are satisfactorily quantified, are summarized in this review. For ease of comparison, all of the antimicrobial potencies are presented in µg/mL and activities of the positive control are also given wherever available.

## 3. Structural Features Which Affect the Antimicrobial Activity of the Compounds

ELF are metabolically versatile organisms that can produce secondary metabolites belonging to different natural product classes. However, by observing the structures of the compounds isolated from ELF categorized above, some common scaffolds leading to distinct antimicrobial properties can be identified. Knowledge of the bioactivities of such chemical scaffolds plays an important role in rational drug discovery and in natural product-related research to make intelligent guesses about the potentials of isolated compounds. The presence of a large pool of data about the potencies of natural compounds or their synthetic or semi-synthetic derivatives with common scaffolds will be helpful in structure–activity relationship (SAR) studies. In order to facilitate such studies, we have summarized the structural scaffolds in Table 2 that can be identified commonly among the antimicrobial compounds isolated from ELF.

Pyrone-related structures can be frequently seen among the compounds listed above. Compounds with the α-Pyrone (**80**) ring arrangement including isocoumarins (**81**) mark the most abundant structural sub-class among all the ELF-derived antimicrobial compounds. Single-ring α-Pyrones such as 3-(2-oxo-2*H*-pyran-6-yl)propanoic acid (**53**), Pericocin C (**56**), and Pericocin D (**57**) showed moderate, specific antifungal activity against *A. niger*, without showing any significant antifungal activity against *C. albicans* or any antibacterial activity [54]. Their antimicrobial properties are significantly different even though **25**, **68** and **70** share the same 6,8-dihydroxy-3-propylisocoumarin skeleton. Even though all three compounds have antifungal potentials, 3-4 saturation of **25** improved its potency and provided an additional antibacterial property. It is evident that the oxidation of the 2-hydroxy group at the 3-propyl side chain, does not cause a considerable effect on their antifungal properties. **67**, **69** and **71**, which consist of a 6,8-dihydroxy-3-methylisocoumarin structure, exhibit quite similar antifungal properties, implying that neither the 3-4 unsaturation nor the 5-hydroxy moiety impacts their pharmacology. By analysing the structures and the IC_50_ values of **70** and **71**, we can conclude that the presence of a 3-methyl substituent group, instead of a 3-(2-hydroxypropyl) group improves the antifungal properties. The introduction of 7-hydroxy-5-methyl substitutions to the isocoumarin scaffold (**43**, **45**) seems to improve its antifungal potential against *C. albicans* [40]. However, when the structures of **43** and **45** are compared, it is evident that the presence of 3-methyl moiety slightly reduces the IC_50_ values than that of 3-(2-hydroxypropyl) group, in contrast to what is observed in **70** and **71**.

γ-Pyrone ring (**82**) is another frequently observed scaffold among the antimicrobial compounds isolated from ELF. Both Carbanarone A (**20**) and Aspergyllone (**59**) are built up of the same skeleton, containing a γ-Pyrone ring [44]. However, they show completely different antimicrobial profiles, owing to the 3-carbamoyl substitution in **20**. Xanthone (**83**) related compounds are another interesting set containing γ-Pyrone rings with antimicrobial potentials. Griseoxanthone C (**8**) and 6-*O*-methylnorlichexanthone (**5**) are mono-methylated forms of norlichexanthone (**10**) [40]. These methylations improve the antibacterial activity against *B. subtilis*; however, that against methicillin-resistant *S. aureus* (MRSA) is diminished. Even though **10** shows weak antifungal activity against *A. fumigatus*, its 3-*O*-methylated form (**8**) is active against *C. albicans*, but not against *A. fumigatus*. Funiculosone (**1**) and Mangrovamide J (**2**) are antibacterial compounds with xanthone-derived structures [39]. **1** displays an improved antibacterial activity, both against *E. coli* and *S. aureus*, owing to the oxidation at the C_1_ position.

**23**, **64** and **65**, having a 3,4-dihydro-1(2*H*)-naphthalenone scaffold (**85**), display moderate antifungal activity against *C. albicans* [45]. **65** has an improved antifungal activity as well as a reduced antibacterial activity, compared to that of **23**, because of the additional 3β-hydroxy group in **65**. Compound **64** has an inverted stereochemistry at the C_3_ position and lacks the 6-hydroxy functionality present in **65**. However, these structural variations seem to cause minimal effect to their antifungal activities.

Compounds **17**, **61** and **63** have an anthraquinone-like structure derived from benzoisochromene (**84**) [29,43]. As **17** is a good antibacterial compound [43], other compounds with similar structures should have been tested to screen their antibacterial potentials; however, only their antifungal properties against *C. albicans* have been tested [29]. Even though compound **60**, which has a Naphthazarin skeleton, lacks a closed pyran ring, it has a similar atomic array to that of **63**. This slight difference in the ring system has not caused a significant impact on their antifungal activities. A similar phenomenon can be observed in Altenusin (**6**) and Alterlactone (**7**), whose only difference is the presence of an opened lactone ring in **6** [40]. Extra structural rigidity and decreased polarity in **7** compared to that in **6**, have not affected their antibacterial activities. However, **7** lacks any antifungal potential against the tested organisms (i.e., *C. albicans* and *A. fumigatus*), unlike **6,** making it a better selective antimicrobial agent.

Another significant group of antimicrobials isolated from ELF is ambuic acid (**26**) and its derivatives [46,50]. 18-*O*-acetylation seems to slightly improve the antibacterial activity against *S. aureus*. However, this has caused mixed effects on their antifungal properties [50]. It is evident that even a slight modification to a distal functional group in the ambuic acid scaffold can drastically change their antimicrobial profiles. However, achieving a generalized conclusion about the effects of the functional groups is difficult with the available data, as these compounds have been tested in different microbial species with large concentration gaps in between two consecutive readings.

## 4. The Way Forward

Endolichenic fungi are, without a doubt, one of the most propitious origins of small molecular natural products with a broad range of therapeutic applications. Many new natural products with different biological activities can be drawn out from these multifaceted organisms. Nevertheless, many factors adversely impact the isolation of efficacious compounds from ELF and they should be understood and overpowered, for the productivity of the process to be enhanced.

One of the major problems is drawing the line between endolichenic microbiota and other strains like mycobiont of the lichen, lichenicolous fungi, airborne and epiphytic microorganisms. Collecting the samples carefully and employing proper surface sterilization techniques can help to remove other abundant microorganisms without bringing about apparent harm to the endolichenic population [3]. One of the well-known, yet rarely discussed drawbacks is the selectively isolating ELF that grows quicker than others. This drawback is evident in most of the processes where several species of microorganisms are isolated and subcultured [36]. Many important ELF and their metabolites may have been left out from observation and analyses due to their cultures growing slowly. Nourishing the ELF in diluted, separated-out cultures, while supplying plenty of time and differing conditions for slow developers to emerge, will aid in overcoming this limitation, at least partly.

Fungi synthesize much of the compounds in minute amounts, therefore separation, determination and characterization of their bioactivities and even their detection becomes challenging [61]. For example, in some situations, even though the solvent extracts of ELF show significant antimicrobial properties, none of the isolated compounds display a similar activity. This could be due to the low percentage abundance of the active compound present in the extract. With novel unknown compounds, mass cultures are required because substantial amounts of afforded products are taken up for spectroscopic techniques during characterization. The numerous advantages of ELF are subdued by the low yields, which limits their applications in industrial scale.

Another main downside of these studies is the lack of ability to prepare the growth media in such a way that the native environment of the lichen is mimicked [36]. Because most useful secondary metabolites are produced as a reaction to stimuli or stress, if the necessary conditions are not fulfilled, the genes responsible for their biosynthesis will be silenced. Differences between laboratory settings and natural surroundings could be unfavourable, yet the same phenomenon may be turned into an advantage. Changes in parameters of the culture such as temperature, oxygen supply, salinity, space, composition and pH of the medium can pave the way to alterations in the metabolome of the ELF, as minute changes can amplify to massive ones [62]. For instance, one of our co-authors has reported a change in the key metabolite of the fungal species *Paraphaeosphaeria quadriseptata* and the isolation of six novel natural products, upon changing the solvent used for the preparation of culture media, from tap water to distilled water. This fascinating change is presumed to be brought about by Cu^2+^ ions present in tapwater in 0.15 ppm levels [63].

OSMAC, which stands for One Strain Many (Active) Compounds, is a technique in which the conditions of the culture media are systematically varied, that can be used to uncover the entire ELF metabolome [62]. Introducing natural conditions into the cultivation medium by using cell lines belonging to the host tissue, co-culturing other symbionts alongside the ELF and exposing the culture to various stress conditions they might face in their natural habitat, will help to override the aforementioned drawbacks and disclose the unseen chemical diversity of ELF. Coculturing bacterial species along with fungal strains is expected to escalate the metabolic diversity of these fungi. For example, co-cultivation of the bacterial species *Streptomyces rapamycinicus* with the fungus *A. fumigatus*, led to the isolation of fumicycline A and B, which are bactericidal in nature [64].

Furthermore, gathering knowledge about as many compounds as possible will aid in creating rich molecular databases that can be used for screening and identifying bioactive metabolites using in silico methods. Once a compound is isolated from an endolichenic fungal species, only a selected number of bioactivity studies are carried out and therefore, some important bioactivities could be overlooked. The genes in the control of the synthesis of compounds and expressing certain properties can be detected by combining environmental genomics with research concerning ELF [65]. Even the genome of a less frequently occurring ELF can be studied with the help of PCR techniques. Once the identification process is complete, the approach of OSMAC can be utilized to improve the fermentation conditions required to generate the greatest amount of the essential metabolite.

It is not practical to investigate all 20,000 lichen species to isolate ELF and yield bioactive compounds [12]. Strobel et al. proposed a procedure for the selection of plants to isolate endophytic fungi with significant bioactivities [66]. As ELF dwell inside lichens, showing similarities to endophytic fungi that reside within plant tissues, Strobel’s postulates can also be adapted to ELF. The postulates to select lichens for bioactivity screening are as follows: (i) lichens with extraordinary biology and more effective adaptations for survival, (ii) lichens with known bioactivities or known to produce bioactive lichen substances, (iii) endemic lichens or ones with an extraordinary evolutionary stability, and (iv) lichens with highly diverse surroundings. Lichens known to have medicinal values are one of the first choices for research, as their symbiotic microbiota are believed to imitate their host and biosynthesize similar or even more biologically active products. Exploring the areas discussed above will provide many undiscovered secrets of nature, and eventually lead to the advancement of therapeutics.

## Figures and Tables

**Figure 1 molecules-26-03901-f001:**
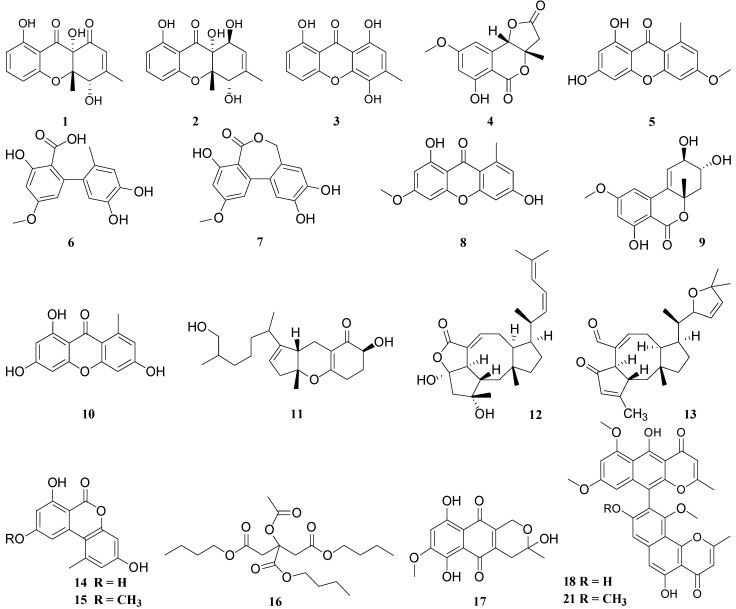
Structures of the antimicrobial compounds isolated from ELF.

**Table 1 molecules-26-03901-t001:** Antimicrobial Compounds Isolated from ELF.

Lichen	Endolichenic Fungi	No.	Compound	Microorganism	Activity (µg/mL)	Positive Control	Activity (µg/mL)	Reference
*Diorygma hieroglyphicum*	*Talaromyces funiculosus*	**1**	Funiculosone	*Escherichia coli*	IC_50_ = 25	-	-	[39]
				*Staphylococcus aureus*	IC_50_ = 58	-	-	
		**2**	Mangrovamide J	*Escherichia coli*	IC_50_ = 65	-	-	
				*Staphylococcus aureus*	IC_50_ = 104	-	-	
		**3**	Ravenilin	*Escherichia coli*	IC_50_ = 23	-	-	
				*Pseudomonas aeruginosa*	IC_50_ = 96	-	-	
				*Staphylococcus aureus*	IC_50_ = 25	-	-	
*Everniastrum* sp.	*Ulocladium* sp.	**4**	6-hydroxy-8-methoxy-3a-methyl-3a,9b-dihydro-3*H*-furo[3,2-c]isochromene-2,5-dione	*Bacillus subtilis*	IC_50_ = 25.0	Gentamicin	IC_50_ < 0.048	[40]
		**5**	6-*O*-methylnorlichexanthone	*Bacillus subtilis*	IC_50_ = 0.39			
		**6**	Altenusin	*Bacillus subtilis*	IC_50_ = 11.3			
		**7**	Alterlactone	*Bacillus subtilis*	IC_50_ = 11.8			
		**8**	Griseoxanthone C	*Bacillus subtilis*	IC_50_ = 0.35			
		**9**	Isoaltenuene	*Bacillus subtilis*	IC_50_ = 14.7			
		**10**	Norlichexanthone	*Bacillus subtilis*	IC_50_ = 0.58			
				Methicillin Resistant *Staphylococcus aureus*.	IC_50_ = 5.4	Vancomycin	IC_50_ < 1.03	
		**11**	Tricycloalternarene 1b	Bacille Calmette-Guérin strain	MIC = 125	-	-	[41]
	*Ulocladium* sp. (CHMCC 5507)	**12**	Ophiobolin P	*Bacillus subtilis*	MIC = 12.6	Gentamicin	MIC = 0.05	[27]
				Methicillin Resistant *Staphylococcus aureus*.	MIC = 25.1	Vancomycin	MIC = 1.0	
		**13**	Ophiobolin T	Bacille Calmette-Guérin strain	MIC = 12.7	Hygromycin	MIC = 0.35	
				*Bacillus subtilis*	MIC = 6.3	Gentamicin	MIC = 0.05	
				Methicillin Resistant *Staphylococcus aureus*	MIC = 12.7	Vancomycin	MIC = 1.0	
*Parmelinella wallichiana*	*Nigrospora sphaerica*	**14**	Alternariol	*Bacillus subtilis*	MIC = 31.2	Amikacin sulfate	MIC = 0.45	[42]
				*Escherichia coli*	MIC = 62.5		MIC = 0.90	
				*Staphylococcus aureus*	MIC = 62.5		MIC = 0.90	
		**15**	Alternariol-9-methyl ether	*Bacillus subtilis*	MIC = 62.5		MIC = 0.45	
				*Pseudomonas* *fluorescens*	MIC = 31.2		MIC = 0.90	
				*Staphylococcus aureus*	MIC = 62.5		MIC = 0.90	
*Parmotrema rampoddense*	*Fusarium* *proliferatum*	**16**	Acetyl tributyl citrate	*Klebsiella pneumoniae*	MIC = 125	-	-	[43]
				*Pseudomonas aeruginosa*	MIC = 125	-	-	
				*Staphylococcus aureus*	MIC = 125	-	-	
		**17**	Fusarubin	*Escherichia coli*	MIC = 1.56	-	-	
				*Pseudomonas aeruginosa*	MIC = 1.56	-	-	
				*Staphylococcus aureus*	MIC = 1.56	-	-	
*Parmotrema ravum*	*Aspergillus niger*	**18**	Asperpyrone A	*Staphylococcus aureus* MTCC 737	IC_50_ = 112	-	-	[44]
		**19**	Aurasperone A	*Dickeya solani* GBBC 1502	IC_50_ = 63	-	-	
				*Listeria innocua* LMG11387	IC_50_ = 141	-	-	
				*Pectobacterium* sp.	IC_50_ = 76	-	-	
				*Pseudomonas aeruginosa* MTCC 424	IC_50_ = 160	-	-	
				*Pseudomonas syringae* pv. Maculicola I11004	IC_50_ = 80	-	-	
				*Staphylococcus aureus* MTCC 737	IC_50_ = 135	-	-	
		**20**	Carbonarone A	*Dickeya solani* GBBC 1502	IC_50_ = 88	-	-	
		**21**	Fonsecinone A	*Escherichia coli* MTCC 443	IC_50_ = 47	-	-	
				*Pseudomonas syringae* pv. Maculicola I11004	IC_50_ = 154	-	-	
				*Staphylococcus aureus* MTCC 738	IC_50_ = 120	-	-	
		**22**	Pyrophen	*Aeromonas hydrophila* ATCC 7966	IC_50_ = 78	-	-	
				*Listeria innocua* LMG11387	IC_50_ = 86	-	-	
				*Micrococcus luteus* DPMB3	IC_50_ = 63	-	-	
*Sticta fuliginosa*	*Xylariaceae* sp. (CR1546C)	**23**	(*R*)-4,6,8-trihydroxy-3,4-dihydro-1(2*H*)-naphthalenone	*Bacillus subtilis*	IC_50_ = 104.2	Streptomycin sulphate	IC_50_ = 5.2	[45]
		**24**	18-*O*-acetylambuic acid	*Staphylococcus aureus* ATCC 6538	IC_50_ = 10.9	Antimicrobial peptide (AMP)		[46]
		**25**	6,8-dihydroxy-(3*R*)-(2-oxopropyl)-3,4-dihydroisocoumarin	*Bacillus subtilis*	IC_50_ = 106.4	Streptomycin sulphate	IC_50_ = 5.2	[45]
		**26**	Ambuic acid	*Staphylococcus aureus* ATCC 6538	IC_50_ = 15.4	Antimicrobial peptide (AMP)		[46]
*Usnea* sp.	*Hypoxylon fuscum*	**27**	16-α-D-mannopyranosyloxyisopimar-7-en-19-oic acid	*Staphylococcus aureus* CGMCC 1.2465	MIC = 46.4	Vancomycin Hydrochloride	MIC = 3.12	[47]
		**28**	8-methoxy-1-naphthyl-β-glucopyranoside	*Staphylococcus aureus* CGMCC 1.2465	MIC = 30.1			
		**29**	Phomol	*Staphylococcus aureus* CGMCC 1.2465	MIC = 21.1			
-	*Coniochaeta* sp.	**30**	Coniothienol A	*Enterococcus faecalis* (CGMCC 1.2535)	IC_50_ = 4.89	Ampicillin	IC_50_ = 2.61	[48]
				*Enterococcus faecium* (CGMCC 1.2025)	IC_50_ = 2.00		IC_50_ = 0.51	
		**31**	Coniothiepinols A	*Enterococcus faecalis* (CGMCC 1.2535)	IC_50_ = 11.51		IC_50_ = 2.61	
				*Enterococcus faecium* (CGMCC 1.2025)	IC_50_ = 3.93		IC_50_ = 0.51	
*Cetraria islandica*	*Myxotrichum* sp.	**32**	Myxodiol A	*Candida albicans* SC 5314	MIC = 128	Fluconazole	MIC = 2	[49]
	*Pestalotiopsis* sp.	**33**	Ambuic acid derivative 1	*Fusarium oxysporum*	MIC = 8	Ketoconazole	MIC = 8	[50]
		**34**	Ambuic acid derivative 2	*Fusarium oxysporum*	MIC = 32		MIC = 8	
		**35**	Ambuic acid derivative 4	*Verticillium dahlia*	MIC = 32		MIC = 1	
		**36**	Ambuic acid derivative 5	*Fusarium gramineum*	MIC = 8		MIC = 8	
				*Fusarium oxysporum*	MIC = 8		MIC = 8	
				*Verticillium dahlia*	MIC = 16		MIC = 1	
		**37**	Ambuic acid derivative 6	*Fusarium gramineum*	MIC = 8		MIC = 8	
		**38**	Ambuic acid derivative 7	*Rhizoctonia solani*	MIC = 32		MIC = 8	
		**39**	Ambuic acid derivative 8	*Rhizoctonia solani*	MIC = 32		MIC = 8	
		**40**	Ambuic acid derivative 9	*Fusarium gramineum*	MIC = 32		MIC = 8	
				*Fusarium oxysporum*	MIC = 16		MIC = 8	
		**41**	Ambuic acid derivative 11	*Fusarium gramineum*	MIC = 32		MIC = 8	
*Cetrelia* sp.	*Aspergillus* sp. CPCC 400810	**42**	Isocoumarindole A	*Candida albicans*	MIC = 32.0	Caspofungin	MIC = 0.03	[51]
*Diorygma hieroglyphicum*	*Talaromyces funiculosus*	**1**	Funiculosone	*Candida albicans*	IC_50_ = 35	-	-	[39]
*Everniastrum* sp.	*Ulocladium* sp.	**43**	7-hydroxy-3-(2-hydroxy-propyl)-5-methyl-isochromen-1-one	*Candida albicans* SC 5314	IC_50_ = 45.4	Amphotericin B	IC_50_ = 1.03	[40]
		**44**	7-hydroxy-3,5-dimethyl-isochromen-1-one	*Candida albicans* SC 5314	IC_50_ = 18.7			
		**6**	Altenusin	*Aspergillus fumigatus*	IC_50_ = 57.5		IC_50_ = 0.74	
		**8**	Griseoxanthone C	*Candida albicans* SC 5314	IC_50_ = 40.6		IC_50_ = 1.03	
		**10**	Norlichexanthone	*Aspergillus fumigatus*	IC_50_ = 43.6		IC_50_ = 0.74	
		**45**	Rubralactone	*Aspergillus fumigatus*	IC_50_ = 63.3		IC_50_ = 0.74	
				*Candida albicans* SC 5314	IC_50_ = 54.7		IC_50_ = 1.03	
*Lethariella zahlbruckner*	*Tolypocladium cylindrosporum*	**46**	Pyridoxatin	*Candida albicans* (Multiple strains)	MIC = 0.5 − 8.0	Fluconazole	MIC = 1.0 − 2.0	[52]
				*Candida glabrata* (Multiple strains)	MIC = 1.0 − 8.0		MIC = 1.0 − 2.0	
				*Candida krusei* (Multiple strains)	MIC = 1.0 − 4.0		MIC = 1.0 − 2.0	
				*Candida tropicalis* CT2	MIC = 32		MIC = 2.0	
*Lobaria quercizans*	*Aspergillus versicolor*	**47**	3,7-dihydroxy-1,9-dimethyldibenzofuran	*Candida albicans*	MIC = 64	Fluconazole	MIC = 2	[53]
		**48**	Cordyol C	*Candida albicans*	MIC = 8			
		**49**	Diorcinol D	*Candida albicans*	MIC = 8			
		**50**	Diorcinol I	*Candida albicans*	MIC = 32			
		**51**	Violaceol I	*Candida albicans*	MIC = 8			
		**52**	Violaceol II	*Candida albicans*	MIC = 8			
*Parmelia* sp.	*Periconia* sp.	**53**	3-(2-oxo-2*H*-pyran-6-yl)propanoic acid	*Aspergillus niger*	MIC = 31	Cycloheximide	MIC < 16	[54]
		**54**	Pericocin A	*Aspergillus niger*	MIC = 31	Cycloheximide	MIC < 16	
		**55**	Pericocin B	*Aspergillus niger*	MIC = 31			
		**56**	Pericocin C	*Aspergillus niger*	MIC = 31			
		**57**	Pericocin D	*Aspergillus niger*	MIC = 31			
		**58**	Pericoterpenoid A	*Aspergillus niger*	MIC = 31			[55]
	*Tolypocladium* sp. (4259a)	**46**	Pyridoxatin	*Candida albicans*	MIC = 0.5	-	-	[56]
*Parmelinella wallichiana*	*Nigrospora sphaerica*	**14**	Alternariol	*Candida albicans*	MIC = 80.0	Ketoconazole	MIC = 1.90	[42]
*Parmotrema ravum*	*Aspergillus niger*	**59**	Aspergyllone	*Candida parapsilosis*	IC_50_ = 52	-	-	[44]
		**19**	Aurasperone A	*Candida krusei* MTCC 9215	IC_50_ = 373	-	-	
		**20**	Carbonarone A	*Candida albicans* MTCC 227	IC_50_ = 103	-	-	
				*Candida krusei* MTCC 9215	IC_50_ = 31	-	-	
		**22**	Pyrophen	*Candida albicans* MTCC 227	IC_50_ = 74	-	-	
				*Candida glabrata*	IC_50_ = 97	-	-	
				*Candida utilis* IHEM 400	IC_50_ = 62	-	-	
*Pseudosyphellaria* sp.	*Biatriospora* sp.	**60**	2-acetonyl-3-methyl-5-hydroxy-7-methoxynaphthazarin	*Candida albicans*	MIC = 64	Fluconazole	MIC = 2	[29]
		**61**	6-deoxy-7-*O*-demethyl-3,4-anhydrofusarubin	*Candida albicans*	MIC = 32			
		**62**	Biatriosporin D	*Candida albicans*	MIC = 16			
		**63**	Biatriosporin K	*Candida albicans*	MIC = 64			
*Sticta fuliginosa*	*Xylariaceae* sp. (CR1546C)	**64**	(3*R*,4*S*)-3,4,8-trihydroxy-3,4-dihydro-1(2*H*)-naphthalenone	*Candida albicans*	IC_50_ = 63.2	Amphotericin B	IC_50_ = 1.3	[45]
		**65**	(3*S*,4*S*)-3,4,6,8-tetrahydroxy-3,4-dihydro-1(2*H*)-naphthalenone	*Candida albicans*	IC_50_ = 67.8			
		**23**	(*R*)-4,6,8-trihydroxy-3,4-dihydro-1(2*H*)-naphthalenone	*Candida albicans*	IC_50_ = 78.2			
		**66**	2,4-dihydroxy-6-(2-oxopropyl)-benzoic acid	*Candida albicans*	IC_50_ = 101.3			
		**67**	5,6,8-trihydroxy-3(*R*)-methyl-3,4-dihydroisocoumarin	*Candida albicans*	IC_50_ = 71.4			
		**68**	6,8-dihydroxy-(3)-(2-oxopropyl)-isocoumarin	*Candida albicans*	IC_50_ = 98.1			
		**25**	6,8-dihydroxy-(3*R*)-(2-oxopropyl)-3,4-dihydroisocoumarin	*Candida albicans*	IC_50_ = 71.2			
		**69**	6,8-dihydroxy-3(*R*)-methyl-3,4-dihydroisocoumarin	*Candida albicans*	IC_50_ = 65.1			
		**70**	6,8-dihydroxy-3-[(2*S*)-2-hydroxypropyl]-isocoumarin	*Candida albicans*	IC_50_ = 99.1			
		**71**	6,8-dihydroxy-3-methylisocoumarin	*Candida albicans*	IC_50_ = 67.2			
*Umbilicaria* sp.	*Floricola striata*	**72**	Floricolin A	*Candida albicans*	MIC = 16	-	-	[57]
		**73**	Floricolin B	*Candida albicans*	MIC = 8	-	-	
		**74**	Floricolin C	*Candida albicans*	MIC = 8	-	-	
		**75**	Floricolin D	*Candida albicans*	MIC = 64	-	-	
		**76**	Terphenyl 2	*Candida albicans*	MIC = 64	-	-	
*Usnea baileyi*	*Xylaria venustula*	**77**	*N*-dodecyldiethanolamine (DDE)	*Candida albicans* NCTC713	MIC = 5.5	-	-	[33,58]
		**78**	Piliformic acid	*Colletotrichum gloeosporioides*	MIC = 625.2	Captan	MIC = 5000	[33,59]
						Difenoconazole	MIC = 8.1	
-	*Coniochaeta* sp.	**31**	Coniothiepinols A	*Fusarium oxysporum* (CGMCC 3.2830)	IC_50_ = 13.12	Carbendazim	IC_50_ = 0.44	[48]
*Parmelinella wallichiana*	*Nigrospora sphaerica*	**14**	Alternariol	Herpes Simplex Virus	IC_50_ = 34.9	-	-	[26]
		**15**	Alternariol-9-methyl ether	Herpes Simplex Virus	IC_50_ = 64.0	-	-	
*Usnea baileyi*	*Xylaria venustula*	**79**	Isoplysin A	*Plasmodium falciparum*	MIC = 0.97	-	-	[33,60]

**Table 2 molecules-26-03901-t002:** Common structural scaffolds among the antimicrobial compounds isolated from ELF.

Scaffold	Compounds
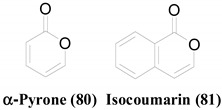	**4**, **9**, **14**, **15**, **25**, **42**, **43**, **44**, **45**, **53**, **54**, **56**, **57**, **66**, **67**, **68**, **69**, **70**, **71**
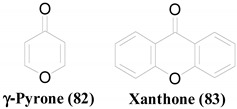	**1**, **2**, **3**, **5**, **8**, **10**, **18**, **19**, **20**, **21**, **30**, **31**, **55**, **59**
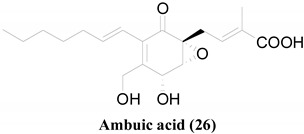	**24**, **26**, **33**, **34**, **35**, **36**, **37**, **38**, **39**, **40**, **41**
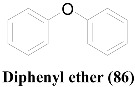	**48**, **49**, **50**, **51**, **52**
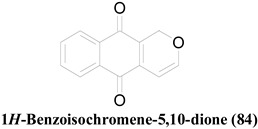	**17**, **61**, **63**
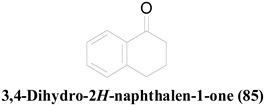	**23**, **64**, **65**

## Data Availability

Not applicable.

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
