# Peer review of "Antimicrobial Compounds Isolated from Endolichenic Fungi: A Review"

_molecules, 2021, doi:10.3390/molecules26133901_

Round 1

Reviewer 1 Report

I read with interest the manuscript.
This manuscript has been well organized as a review for endolichenic fungi and the antimicrobial compounds produced from them. 
This manuscript will be suitable for publishing once the minor changes indicated below are made.

Minor comment

- In Figure 1, the expression of the methyl group in the chemical structure is not uniform. The expression of the methyl group of compounds 4,5,6,7,8,9,10,18,19,20,22,43,44,45 needs to be unified with compounds 55, 60, 62 and 63. 

- In Figure 1, the font corresponding to compound 47-51 is different from other compound numbers. The font needs to be unified. 

- Compound 47 has only the Man mark, and no specific structure is shown.

- Compounds 49 and 50 have also only the Prenyl mark, and no specific structure is shown.

- The wavy side chain representation between ketone and olefin in compounds 78 is not clearly visible. 

Author Response

the reviewers comment are attached as a separate file

Reviewer 2 Report

The review presented focuses specifically on natural products isolated in endolichenic fungi. Overall, the review is well written, very informative and well laid out. I personally liked the tables and structures indicated. The authors have presented the state of the art very professionally and I am sure this review will be a valuable source for many scientists.

I have only few minor suggestions:

Line 15: I would substitute “antibiotics”, already used, with “therapeutics”;

Lines 193 to 218 the formatting is different: compound numbers are not bold and species names are not italicized. Please Correct.

Line 269: Please change to “A. fumigatus”;

Line 273: “in silico” should be italicized;

Line 286: “…screening are as follows:” and you need a comma after “stability” (lines 288-289)

References: overall it is well formatted but the titles are not always in the same style. Some titles have the first letter of the word capitalized. Please be consistent.

Author Response

comments are attached as a separate file
